# Association Study of a Comprehensive Panel of Neuropeptide-Related Polymorphisms Suggest Potential Roles in Verbal Learning and Memory

**DOI:** 10.3390/genes15010030

**Published:** 2023-12-24

**Authors:** Nesli Avgan, Heidi G. Sutherland, Rod A. Lea, Larisa M. Haupt, David H. K. Shum, Lyn R. Griffiths

**Affiliations:** 1Genomics Research Centre, Centre for Genomics and Personalised Health, School of Biomedical Sciences, Queensland University of Technology (QUT), 60 Musk Ave., Kelvin Grove, QLD 4059, Australia; avgan.nesli@gmail.com (N.A.); heidi.sutherland@qut.edu.au (H.G.S.); rodney.lea@qut.edu.au (R.A.L.); 2Stem Cell and Neurogenesis Group, Genomics Research Centre, Centre for Genomics and Personalised Health, School of Biomedical Sciences, Queensland University of Technology (QUT), 60 Musk Ave., Kelvin Grove, QLD 4059, Australia; larisa.haupt@qut.edu.au; 3Centre for Biomedical Technologies, Queensland University of Technology (QUT), 60 Musk Ave., Kelvin Grove, QLD 4059, Australia; 4ARC Training Centre for Cell and Tissue Engineering Technologies, Queensland University of Technology (QUT), Kelvin Grove, QLD 4059, Australia; 5Max Planck Queensland Centre for the Materials Science of Extracellular Matrices, Queensland University of Technology (QUT), Kelvin Grove, QLD 4059, Australia; 6Department of Rehabilitation Sciences, The Hong Kong Polytechnic University, Hong Kong, China; david.shum@polyu.edu.hk

**Keywords:** neuropeptide, neuropeptide receptors, cognition, learning, verbal memory, HVLT-R, genetic polymorphism

## Abstract

Neuropeptides are mostly expressed in regions of the brain responsible for learning and memory and are centrally involved in cognitive pathways. The majority of neuropeptide research has been performed in animal models; with acknowledged differences between species, more research into the role of neuropeptides in humans is necessary to understand their contribution to higher cognitive function. In this study, we investigated the influence of genetic polymorphisms in neuropeptide genes on verbal learning and memory. Variants in genes encoding neuropeptides and neuropeptide receptors were tested for association with learning and memory measures using the Hopkins Verbal Learning Test—Revised (HVLT-R) in a healthy cohort of individuals (n = 597). The HVLT-R is a widely used task for verbal learning and memory assessment and provides five sub-scores: recall, delay, learning, retention, and discrimination. To determine the effect of candidate variants on learning and memory performance, genetic association analyses were performed for each HVLT-R sub-score with over 1300 genetic variants from 124 neuropeptide and neuropeptide receptor genes, genotyped on Illumina OmniExpress BeadChip arrays. This targeted analysis revealed numerous suggestive associations between HVLT-R test scores and neuropeptide and neuropeptide receptor gene variants; candidates include the *SCG5*, *IGFR1*, *GALR1*, *OXTR*, *CCK,* and *VIPR1* genes. Further characterization of these genes and their variants will improve our understanding of the genetic contribution to learning and memory and provide insight into the importance of the neuropeptide network in humans.

## 1. Introduction

Cognition and behaviour are the foremost important outputs of human brain functioning and are underpinned by memory functions mediated by a set of neural mechanisms, which allow information to be encoded, consolidated, retained, and retrieved. Memory provides us with a sense of self that links our past, present, and future and is important for individual, social, educational, and vocational functioning as exemplified by the problems experienced by people with memory impairment or amnesic syndromes due to brain injuries [1].

Different types of memory are mediated by different parts of the brain, although they may interact and partly overlap [2]. Implicit memory refers to the unconscious facilitation of recall or recognition by previously encountered information and relies on the basal ganglia and cerebellum, while explicit memory refers to the ability to recall or recognise information consciously and is associated with the hippocampus, the neocortex, and the amygdala. Most developed memory tests measure explicit memory, which is subdivided into semantic (knowledge of facts) and episodic (recall or recognition of past events and experiences) [3]. Furthermore, there are three subtypes of episodic memory: short-term memory (STM), long-term memory (LTM), and working memory (WM). STM involves processes that last for only a few seconds or minutes and involves hippocampal regions, while LTM entails processes that involve gating of memories formed in the hippocampus through a thalamo-cortical circuit for longer-term cortical storage [4]. WM refers to the ability to hold and actively process information immediately after its presentation and is closely linked to the prefrontal cortex (PFC). Verbal and visuospatial information is also processed in separate domains; verbal tasks have been shown to preferentially involve activity in a network of regions including the PFC and along the inferior prefrontal gyrus, while activity in the occipital–temporal cortex correlates with processing of visual information [5].

Cognitive and behavioural functions are closely entwined and are regulated with overlapping neuronal networks and dependent on chemical neurotransmission. Neurotransmitters can be classified into three categories: fast signalling (including the excitatory amino acids and catecholamines), slow modulatory, and neuropeptides [6]. Neuropeptides are 3-to-100-amino-acid-long hormone-like signalling molecules produced by neurons and some other tissues and are important for communication between neurons and other neural cell types as well as non-neuronal target cells [7]. It is important to note that the number of known neuropeptides far exceeds that of classical neurotransmitters. Due to the slow and more diffuse release of neuropeptides, they are usually considered to be second messengers and have neurotrophic and neuromodulatory effects. Previous studies have shown the involvement of neuropeptides in numerous neuroendocrine and behavioural processes including metabolism, stress control, learning, and memory [8,9,10].

Neuropeptides localize in regions involved in cognition, learning, and memory in the brain and have been characterized as regulators of these pathways [11,12]. To date, research into the function of neuropeptides in learning and memory has mostly been conducted in animal models, with several neuropeptide and receptor families, including the somatostatins, tachykinin, galanin, and vasoactive intestinal peptide (VIP), linked with cognition [12]. For example, VIP and its receptors have been implicated in learning and memory processes in mice through investigating gene knockouts [13] and pharmacological interventions which demonstrate a role in synaptic transmission [14]. Although there is conservation of the neuropeptide network across mammals [15], studying genetic variation in neuropeptide and neuropeptide receptor genes may allow further dissection and characterization of their roles in human cognitive and behavioural functions. Twin studies have demonstrated that specific cognitive and memory traits are genetically influenced, with heritability estimates ranging from 30 to 60% [16,17,18]. Genetic polymorphisms, including single-nucleotide polymorphisms (SNPs) are used in association studies to identify genetic loci that influence human traits and disorders, including complex polygenic cognitive and memory phenotypes [18].

Neuropeptide and neuropeptide receptor gene polymorphisms have been investigated in various human behavioural disorders (e.g., sleep, feeding, and obesity) [19] as well as social and neuropsychological traits [20]. There has been much interest in the oxytocin/vasopressin neuropeptide family and oxytocin receptor gene (*OXTR*); e.g., the common variant rs53576 in *OXTR* has been investigated in personality traits and behaviour [21,22,23,24]. In 2015, Li et al. conducted a meta-analysis of *OXTR* rs53576 and reported a positive association between the SNP and general sociality but not close relationships (parent–child and/or romantic relationship) or depression [25]. SNPs in *OXTR* have also been studied in relation to autism and intelligence with SNPs and haplotypes in the *OXTR* gene identified to confer risk for autism spectrum disorders [26]. In animal studies, oxytocin and vasopressin have been shown to be associated with learning and memory [27], and in humans, shorter microsatellite repeat polymorphisms in the arginine vasopressin receptor 1A (*AVPR1A*) gene promoter have been associated with poorer verbal memory performance via modulation of hippocampal structures and connectivity with the thalamus [28]. Neuropeptide Y (NPY) Y_2_ receptor engagement was shown to be required for learning and memory processing in a mouse knockout model [29]. In humans, the *NPY* -485T promoter allele was shown to be associated with schizophrenia susceptibility due to a decreased level of neuropeptide in the brain [30]; it is worth noting that memory defects are common in schizophrenia.

Although several studies have investigated specific genetic polymorphisms in neuropeptide and neuropeptide receptor genes in relation to social behaviour, learning, and memory, comprehensive studies of neuropeptide-related genetic variants have not been conducted. Here, our aim was to explore associations between an extensive list of neuropeptide and neuropeptide receptor gene variants and verbal learning and memory assessed using the Hopkins Verbal Learning Test—Revised (HVLT-R) in a healthy Australian cohort. The HVLT-R is a widely accepted neurocognitive assessment of verbal learning and memory with measures for five sub-components: recall, delay, learning, retention, and recognition discrimination [31]. In this study, we used neuropeptide databases, http://www.neuropeptides.nl/ (accessed on 10 January 2017) [7] and NeuroPedia [32], to compile a list of more than 100 human genes encoding classical and candidate neuropeptide genes and their receptors. From these, we identified >1300 SNPs present on Illumina^®^ Human OmniExpress-24 BeadChip arrays (Illumina, Inc., San Diego, CA, USA) and tested for genetic associations with the measures of verbal learning and memory assessed by the HVLT-R sub-components in a healthy population cohort.

## 2. Experimental Procedures

### 2.1. Participants

Participants in this study were individuals recruited through advertisements from the Brisbane and Gold Coast areas of South-East Queensland in Australia. Participation was excluded for individuals with a history of psychiatric disorder or head injury to maintain a representative sample of cognitive and memory ability without additional complications, although it should be noted that other neurological disorders that may impact some memory functions, e.g., autism spectrum disorder, were not specifically screened for. Written informed consent was provided by all participants prior to any study activities. The study was approved by the Griffith University (MSC/01/09/HREC) and Queensland University of Technology Human Research and Ethics (1300000486) Committees.

### 2.2. Phenotyping

All participants were assessed individually and under the same conditions, in a quiet and well-lit room, by the same examiner.

#### 2.2.1. Learning and Memory

Verbal learning and memory abilities of the participants were determined using the HVLT-R test [31]. The test is based on three trials of free recall of 12 items followed by a 20 to 30 min delayed free recall and subsequent yes and no recognition from a semantically categorized set of words (n = 24). The HVLT-R evaluates learning and memory using 5 sub-components: recall, delay, learning, retention, and recognition discrimination, which are calculated as follows.
$${Recall}^{HVLT-R}=Trial\;1+Trial\;2+Trial\;3$$
$${Delay}^{HVLT-R}=Trial\;4$$
$${Learning}^{HVLT-R}=max(Trail\;2,\;Trail\;3)-Trial\;1$$
$${Retention}^{HVLT-R}=Trial\;4\;/\;max(Trial\;1,\;Trial\;2,\;Trial\;3)$$
$${Discrimination}^{HVLT-R}=True\;Positive\;Recognition-False\;Positive\;Recognition$$

#### 2.2.2. IQ

The intelligence quotient (IQ) was measured using subsets of the Wechsler Abbreviated Scale of Intelligence (WASI) IQ test. The WASI-IQ test was created in 1955 by David Wechsler and is a well-established IQ test for measuring adult intelligence [33]. The vocabulary and matrix reasoning subsets of WASI were completed in this study to estimate the IQ of participants and was used as one of the covariates in the analysis.

### 2.3. Sample Collection and DNA Extraction

Saliva samples were collected from the participants immediately after completion of the memory tests using Oragene^®^ DNA Self Collection kits (DNA Genotek Inc., Ottawa, ON, Canada). DNA was extracted from whole saliva samples by using the kit and protocol of the same manufacturer.

### 2.4. Genotyping and Gene Selection

Saliva DNA samples were genotyped on Illumina^®^ Human OmniExpress-24 BeadChip arrays (Illumina, Inc., San Diego, CA, USA) and genotype data for 597 individuals were available after quality control (QC). Genes encoding neuropeptides were identified using two neuropeptide databases. A total of 93 genes were obtained using NeuroPedia [32] and nine additional genes incorporated from the www.neuropeptides.nl database (accessed on 10 January 2017) [7]. Furthermore, 53 neuropeptide receptor genes from the Illumina OmniExpress 24 v1.1 annotation file were included in the gene set (Appendix A). From this gene set, we compiled a list of SNPs from the exonic and intronic regions of the loci of interest and extracted the genotype data for the memory cohort. We failed to locate SNPs from 19 of the neuropeptide genes that had been identified from the databases in our genotype data and SNPs from 12 genes did not pass genotyping quality thresholds. Therefore, the final SNP list comprised 1306 SNPs located in 124 neuropeptide and receptor genes which are shown in Table 1.

### 2.5. Statistical Analysis

The R Program for Statistical Computing (v3.3.3) [34] was used to complete the descriptive statistics of the learning and memory phenotypes. Verbal learning and memory scores and IQ were investigated for their correlation using Pearson’s r test. Population structure was studied using The R Program and KING (v1.9), a population structure inference tool [35], and principal component analysis (PCA) was undertaken to consider population structure. Self-reported ancestry did not cluster together, indicating an unrecognised mixed background for various individuals, and supported the use of principal components (PCs) instead of self-reported ethnicity in analyses.

QC of the genotype data and association analyses were conducted using PLINK (v1.09) [36]. QC thresholds for the analysis were determined as follows: minor allele frequency (MAF) and Hardy Weinberg equilibrium (HWE) set to higher than 0.01 and 0.001, respectively. QC was completed for the selected set of markers and SNPs located in the selected genes that passed thresholds were included in the study. Generalized linear model analysis was carried out for each memory phenotype individually to identify correlations. Age, sex, IQ, PC1, and PC2 were considered as covariates in the analysis. Association analyses were performed for the SNPs within genes of interest from the Illumina OmniExpress BeadChip assays to estimate genotypic effects on learning and memory status. To eliminate type I error due to multiple testing, we calculated the suggestive and significance *p-*value thresholds using the Genetic type 1 Error Calculator (GEC) (v0.2) [37]. A suggestive *p-*value threshold was set to 0.00113, while the significance *p-*value was 0.0000564.

## 3. Results

### 3.1. Demographics and Phenotype Analysis

A cohort of healthy participants recruited from the Brisbane and Gold Coast areas of Australia undertook a battery of memory tests, including the HVLT-R to assess verbal learning and memory, and donated a saliva sample to extract DNA for genetic analysis. Statistical analysis was performed for the 597 participants for whom genotyping data were obtained and the demographics of this memory cohort are presented in Table 2. Two-thirds of the cohort were female (71%) with an age range of 16 to 65 years old (M = 20, SD = 8.57). The majority of the cohort identified as Caucasian (75%). Due to the large number of individuals with other ethnicities (n = 151, mostly Asian), population structure was studied using PCA, and the first two PCs were added into analyses as a covariate, along with age and gender.

IQ was normally distributed in our cohort and ranged from 78 to 138 (μ = 108.1). In this study, verbal learning and memory performance was measured using the HVLT-R using five sub-components: recall, delay, learning, retention, and discrimination. When we further examined these sub-components, we found correlations between *recall^HVLT-R^* and *delay^HVLT-R^* measures (0.64) as well as between *delay^HVLT-R^* and *retention^HVLT-R^* measures (0.72). Analysis of HVLT-R subtest scores (recall, delay, learning, retention, discrimination) by age using Pearson’s correlation showed a moderate positive correlation between score and age (r = 0.2, *p* < 0.001), with no other significant correlations observed. Analysis by sex using an independent T-test showed that, in this cohort, males had higher average IQ scores compared to females (*p* < 0.001), while comparisons for HVLT-R scores were all non-significant. Finally, analysis by ethnic group (Caucasian versus non-Caucasian) using an independent-samples T-test showed higher average *recall^HVLT-R^*, *delay^HVLT-R^*, and IQ scores, compared to non-Caucasians (*p* = 0.002, *p* = 0.005, and *p* < 0.001, respectively), while other comparisons were non-significant. Significant results may reflect the relatively small sample sizes for some of the groups and could suggest some impact of English as a second language on some of the verbal memory scores. The results justify adding age, sex, IQ, and PCs 1 and 2 (accounting for population structure) as covariates while subsequent analyses were performed on the entire cohort to maintain statistical power.

### 3.2. Genotype Association

We compiled a comprehensive list of 80 human neuropeptide and 44 neuropeptide receptor genes from databases (Table 1) and identified SNPs in their exonic and intronic regions that were present on Illumina^®^ Human OmniExpress-24 BeadChip arrays. We extracted genotypes for these SNPs from the microarray data of the 597 individuals of the memory cohort. Due to the selected quality control thresholds (minor allele frequency (MAF) > 0.01; Hardy Weinberg equilibrium (HWE) > 0.001), eight neuropeptide genes and four neuropeptide receptors were excluded from the analysis. Association analyses with the test scores from each of the HVLT-R sub-components were performed for the 1306 remaining SNPs using a generalized linear model (GLM). As our memory cohort comprised mostly females (71%), young adults (78% were 16 to 25 years old), and Caucasians (75%) (Table 2), we included age, gender, and population structure (PC1 and PC2) as well as IQ to the analysis as covariates in order to minimize their effect on the study and to focus on the association of the loci of interest and learning and memory. To consider and eliminate type I error due to multiple testing, the *p-*value thresholds were calculated and adjusted to be able to declare the significant SNPs. To calculate our adjusted *p-*value thresholds, SNPs in linkage disequilibrium (LD) were considered and after estimating the effective number of independent markers (n = 887), we set the significance threshold to 0.0000564 and a suggestive *p-*value as 0.00113.

While 298 markers were below an α level of 0.05 in one phenotype or more, none of the markers in the loci of interest were significantly associated with verbal learning and memory after the multiple testing corrections (Appendix A). However, nine SNPs were found to be suggestively associated with HVLT-R phenotypes, with rs1997317 in the Secretogranin V gene (*SCG5*) approaching significance with the *delay^HVLT-R^* component (Table 3). Box plots showing the allelic distributions of the suggestive SNPs presented in Table 3 are illustrated in Figure 1.

We then used the GTEx (Genotype-Tissue Expression) expression quantitative trait loci (eQTL) Browser (http://www.gtexportal.org/home/) to further investigate eQTLs of the SNPs potentially linked to learning and memory traits in Table 3. rs11634086 is a significant eQTL for *SCG5* in testis (*p* = 1.47 × 10^−6^), but it is also an eQTL in various human brain regions including the cortex (*p* = 6.5 × 10^−5^). In addition to the pancreas and pituitary, *SCG5* shows high expression levels in brain-related tissues (Figure 2), which may support the supposition that the SNPs we have identified may impact or are in high LD with SNPs that affect memory performance.

rs2684788 is an eQTL for *IGF1R* in brain tissues, including the cerebellum (*p* = 5.1 × 10^−15^) and hippocampus (2.6 × 10^−4^), and shows expression in these regions as well as high expression in arteries and thyroid tissue. *GALR1* has higher expression only in the pituitary according to GTEx. The SNP rs8192472 is reported as an eQTL in GTEx for *CCK* in the traverse colon (*p* = 4.3 × 10^−5^), with *CCK* being highly expressed in some tissue regions of the brain (Figure 3). rs437876 is an eQTL for *VIPR1* in whole blood (1.4 × 10^−7^), with some signal in brain regions such as the cortex (*p* = 2.9 × 10^−3^) and amygdala (*p* = 0.03). The other suggestively associated SNPs in Table 3 were not present in GTEx. Both *VIPR1* and *OXTR* are expressed in some brain regions, including areas important for cognition, but *VIPR1* is most highly expressed in the lungs and *OXTR* in breast tissue according to GTEx.

## 4. Discussion

In the present study, we performed targeted analysis of SNPs in neuropeptide- and neuropeptide receptor-encoding genes, to test for associations between genotypes and verbal learning and memory performances as assessed by the HVLT-R, in a healthy Australian cohort. We found numerous SNPs in various neuropeptide and neuropeptide receptor genes with a *p*-value < 0.05 (Appendix A). However, after multiple-testing correction and adjustment to the significance threshold (0.0000564), no SNPs were identified as significant. Nevertheless, suggestive associations (below the suggestive *p-*value of 0.00113) remained for nine SNPs in six neuropeptide and neuropeptide receptor genes (*SCG5, IGF1R, GALR1, OXTR, CCK, VIPR1*) with specific HVLT-R components of recall, delay, learning, retention, and discrimination (Table 3).

Three suggestively associated SNPs were identified in the *SCG5* gene that encodes Secretogranin V, belonging to the chromogranin and secretogranin family of proteins; rs11634086 and rs10519737 (both located in intron 2) were associated with the *recall^HVLT-R^* score and rs1997317 (located in the final intron) was associated with the *delay^HVLT-R^* score. Minor alleles of these SNPs were positively associated with higher memory performance in the *recall^HVLT-R^* and *delay^HVLT-R^* sub-components. The recall and delay sub-components are the only memory phenotypes assessed by the test, with *recall^HVLT-R^* assessing immediate verbal memory, and longer-term verbal memory assessed by the *delay^HVLT-R^*, using the same word list but measured after a 20 to 30 min interval. This study presents the first suggestion that polymorphisms in *SCG5* may be associated with human memory and is supported by in silico analysis of *SCG5* expression showing high levels in the brain and its sub-regions important in learning and memory (Figure 2). While these SNPs have not been previously associated with any phenotypes, Cao-Lei et al. previously reported a behavioural association for *SCG5*, showing that DNA methylation changes in *SCG5* are highly correlated with maternal objective stress [38].

The insulin-like growth factor 1 receptor (*IGFR1*) is a member of the insulin gene family. Two SNPs at this gene locus, rs867431 and rs2684788, showed associations with the *learning^HVLT-R^* and *retention^HVLT-R^* scores, respectively. These SNPs have not been previously linked to human behaviour or learning and memory. rs2684788 is a 3′UTR SNP and was previously reported to be associated with susceptibility to idiopathic short stature [39]. The other SNP that showed correlation with the *retention^HVLT-R^* score was the galanin receptor 1 (*GALR1*) polymorphism rs9959924. *GALR1* belongs to the galanin neuropeptide family; galanin has been studied in Alzheimer’s disease, epilepsy, and memory consolidation studies [40,41,42,43] and is known for its potential role in learning and memory [11]. The *GALR1* SNP rs9959924 we identified to be potentially associated with verbal retention was previously studied in relation to smoking and nicotine dependence and found to be significantly associated with smoking quantity [44]. While not directly related to memory, it suggests the SNP may have a functional effect.

SNPs rs237888 (*OXTR*), rs8192472 (*CCK*), and rs437876 (*VIPR1*) showed suggestive associations with the *descrimination^HVLT-R^* score, an index obtained by subtracting the number of objects recognized falsely from a list from the number of true positives. *OXTR* is an oxytocin receptor gene in which a number of polymorphisms have been widely studied in autism, cognition, and behaviour [26,45,46,47]. While we did not find high LD between rs237888 and these other *OXTR* SNPs reported in the literature, however, *OXTR* SNP rs237888 has previously been found to be associated with recognition and decision making [48], consistent with our findings. The cholecystokinin gene (*CCK*) has been studied in relation to anxiety and cognition [49] and more recently has been associated with body mass index, similar to other neuropeptides that have been found to regulate eating behaviour and energy homeostasis [50]. Furthermore, in murine models, CCK has been found to modulate hippocampal long-term potentiation (LTP) and spatial learning and memory by facilitating neuroplasticity of hippocampal CA3-CA1 synapses [51].

*VIPR1*, vasoactive intestinal polypeptide receptor 1 gene, encodes the receptor for *VIP,* which is known to play a modulatory role in learning and memory [52]. VIP is a neuropeptide released from dentate gyrus interneurons and enhances the proliferative and proneurogenic effect of microglia via IL-4 release by the VPAC1 receptor [53]. Furthermore, Cunha-Reis et al. studied long-term depression, focusing on *VIPR1* for possible pharmacological targets for treatment of cognitive dysfunction [54]. This supports our finding that *VIPR1* polymorphisms may play a role in learning and memory. Although *VIPR1* shows only moderate expression in tissues related to the brain or cognition in GTEx, *VIP* and its receptor gene *VIPR1* are expressed in the postnatal dentate gyrus where they direct neural stem/precursor cells towards a neuronal fate [52].

There are a number of limitations in our study. Although we identified suggestive associations for neuropeptide-related genes in learning and memory phenotypes, none were significant. The relatively small sample size may have contributed to SNPs failing to pass the significance threshold. Furthermore, we did not impute our genotype data due to the considerable proportion of non-Caucasian individuals in the cohort; imputation may have increased power to detect possible associated variants and would have provided a higher resolution of association signals. Another limitation is that the SNPs were not validated in an independent cohort. Since the HVLT-R is a widely used assessment of learning and memory, it would be of interest to investigate these SNPs and genes in other cohorts phenotyped using either this specific test or other verbal learning and memory measures. We also only included SNPs that were in the coding and intronic regions of the neuropeptide and neuropeptide receptor genes in the study. SNPs in the upstream or downstream regulatory regions of these genes may also affect their expression levels and may, therefore, also be relevant for future studies. Finally, this was a targeted study and genome-wide approaches are necessary for more comprehensive association testing, although such approaches require larger sample sizes, particularly if cohorts or phenotyping is heterogeneous. Notably, the well-known Alzheimer’s disease *APOE*/*APOC1*/*TOMM40* region has been associated with verbal memory phenotypes in a number of studies [55,56,57], and a large meta-analysis additionally identified significant signals at *CDH18*, *NT5DC2*, *STAB1*, *ITIH1*, *ITIH4*, and *PBRM1* and implicated synaptic and neurodevelopmental genes [57].

## 5. Conclusions

Our study is the first to examine verbal learning and memory in a healthy population assessed using the HVLT-R with respect to a comprehensive list of neuropeptide and neuropeptide receptor gene variants. In summary, we found a number of suggestive associations between SNPs in genes related to neuropeptide networks with verbal learning and memory. Our findings establish new potential targets for future studies and enhance our knowledge on the importance of neuropeptides in learning and memory. Interestingly, similar to work in animal studies, treatment of patients with memory deficits (e.g., with schizophrenia) with exogenous neuropeptides such as arginine vasopressin has been shown to have potential therapeutic benefits for cognitive and memory functions [58]. Further work investigating the role of neuropeptide and neuropeptide receptor genes in learning and memory is warranted.

## Figures and Tables

**Figure 1 genes-15-00030-f001:**
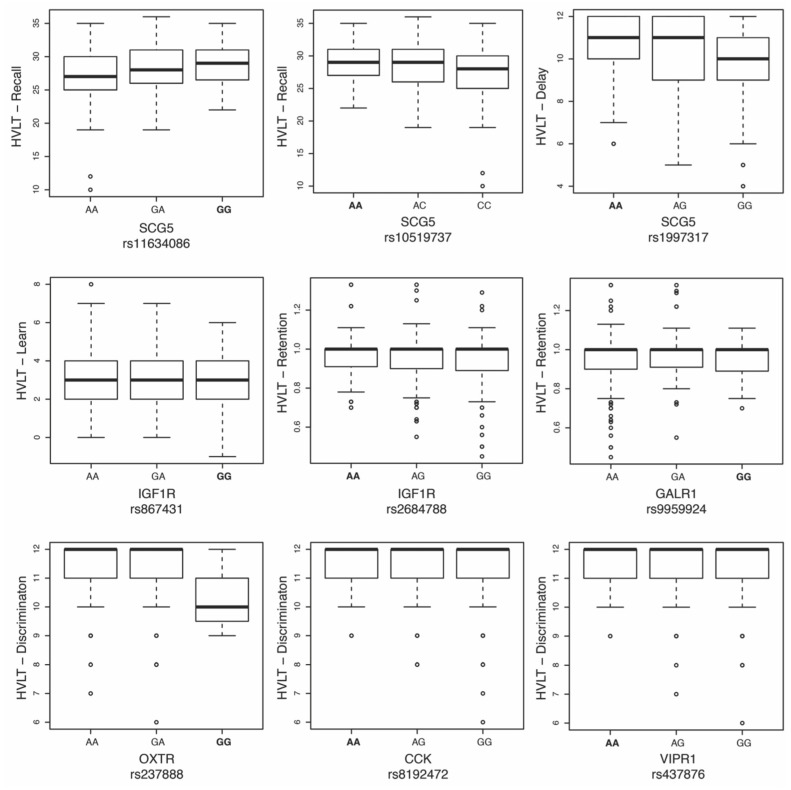
Hopkins Verbal Learning Test—Revised (HVLT-R) phenotypes and allelic distributions of suggestively associated SNPs from neuropeptide and neuropeptide receptor genes. Genotypes homozygous for the minor allele are bolded. Circles represent values of outlier samples.

**Figure 2 genes-15-00030-f002:**
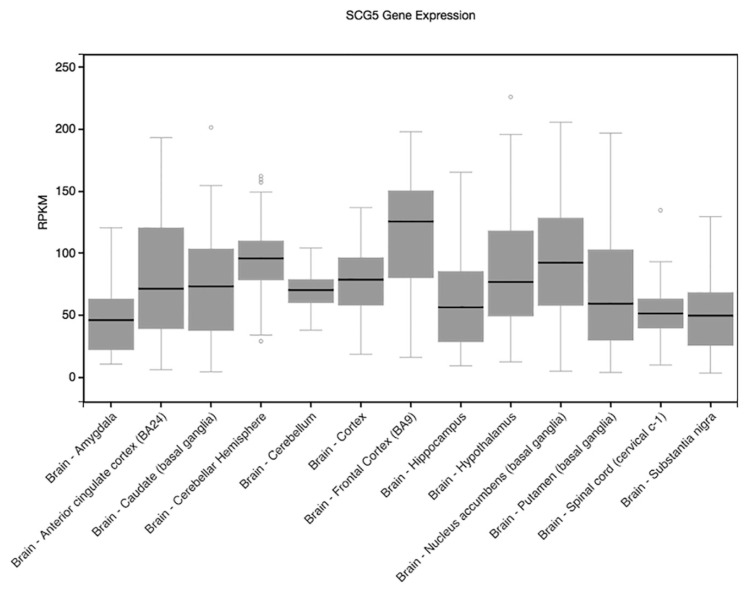
Genotype-Tissue Expression (GTEx) *SCG5* gene expression illustrated across brain regions. (RPKM: reads per kilo base transcript per million). Circles represent values of outlier samples.

**Figure 3 genes-15-00030-f003:**
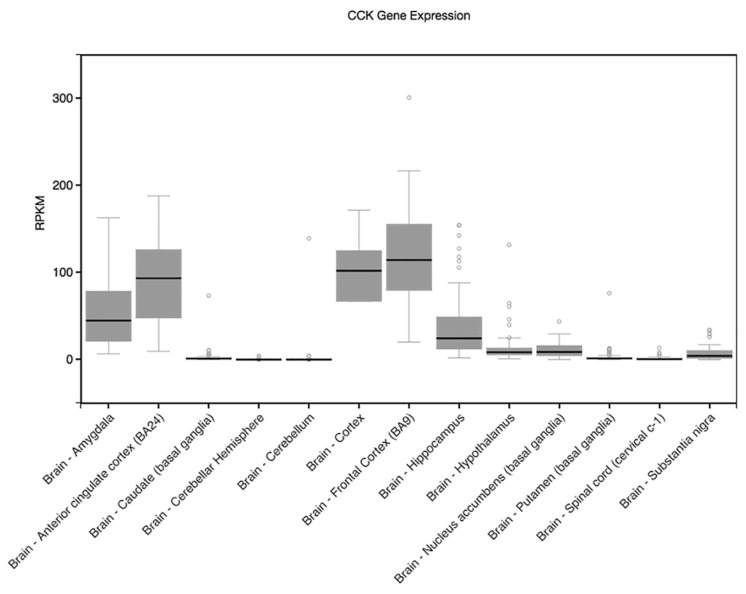
GTEx *CCK* gene expression illustrated across brain regions. (RPKM: reads per kilo base transcript per million). Circles represent values of outlier samples.

**Table 1 genes-15-00030-t001:** Overview of neuropeptide-encoding genes and neuropeptide receptor genes that were analysed in this study.

Gene Family	Gene	Gene Name
Acyl-CoA Binding Protein	*DBI*	diazepam binding inhibitor
Adipose neuropeptides	*ADIPOQ*	adiponectin
	*LEP*	leptin
	*LEPR*	leptin receptor
	*NUCB2*	nucleoblindin 2
	*RETN*	resistin
	*RETNLB*	resistin-like beta
	*UBL5*	ubiquitin-like 5
AVIT (prokineticin)	*PROK1*	prokineticin 1
	*PROK2*	prokineticin 2
Bombesin-like peptides	*GRP*	gastrin-releasing peptide
	*NMB*	neuromedin B
	*NMBR*	neuromedin B receptor
Calcitonin	*CALCA*	calcitonin-related polypeptide alpha
	*CALCB*	calcitonin-related polypeptide alpha
	*IAPP*	islet amyloid peptide
Cholecystokinin/gastrin	*CCK*	cholecystokinin
	*CCKAR*	cholecystokinin A receptor
	*CCKBR*	cholecystokinin B receptor
Cerebellins	*CBLN1*	cerebellin 1 precursor
	*CBLN2*	cerebellin 2 precursor
	*CBLN3*	cerebellin 3 precursor
	*CBLN4*	cerebellin 4 precursor
Corticotropin-releasing hormone	*CRHBP*	corticotropin-releasing hormone binding protein
	*CRHR1*	corticotropin-releasing hormone receptor 1
	*CRHR2*	corticotropin-releasing hormone receptor 2
Corticotropin-releasing factor	*UCN2*	urocortin 2
	*UCN3*	urocortin 3
	*UTS2*	urotensin 2
Endothelin	*EDN1*	endothelin 1
	*EDN2*	endothelin 2
	*EDN3*	endothelin 3
F- and Y- amides	*NPFF*	neuropeptide FF-amide peptide precursor
	*NPFFR1*	neuropeptide FF receptor 1
	*NPFFR2*	neuropeptide FF receptor 1
	*NPY*	neuropeptide Y
	*NPY1R*	neuropeptide Y receptor Y1
	*NPY2R*	neuropeptide Y receptor Y2
	*NPY5R*	neuropeptide Y receptor Y5
	*PPY*	pancreatic polypeptide
	*PYY*	peptide YY
	*QRFP*	pyroglutamylated RFamide peptide
	*QRFPR*	pyroglutamylated RFamide peptide receptor
Galanin	*GAL*	galanin and GMAP prepropeptide
	*GALP*	galanin-like peptide
	*GALR1*	galanin receptor 1
Glucagon/secretin	*ADCYAP1*	adenylate cyclase activating polypeptide 1
	*ADCYAP1R1*	ADCYAP receptor type 1
	*GCG*	glucagon
	*GHRH*	growth hormone-releasing hormone
	*GHRHR*	growth hormone-releasing hormone receptor
	*GIP*	gastric inhibitory polypeptide
	*GIPR*	gastric inhibitory polypeptide receptor
	*SCTR*	secretin receptor
	*VIP*	vasoactive intestinal peptide
	*VIPR1*	vasoactive intestinal peptide receptor 1
	*VIPR2*	vasoactive intestinal peptide receptor 2
Gonadotropin-releasing hormone	*GNRH1*	gonadotropin-releasing hormone 1
	*GNRH2*	gonadotropin-releasing hormone 2
	*APLNR*	apelin receptor
	*HCRTR1*	hypocretin receptor 1
	*HCRTR2*	hypocretin receptor 2
Granins	*CHGA*	chromogranin A
	*CHGB*	chromogranin B
	*SCG2*	secretogranin II
	*SCG3*	secretogranin III
	*SCG5*	secretogranin V
Insulin	*IGF1*	insulin-like growth factor 1
	*IGF1R*	insulin-like growth factor 1 receptor
	*IGF2R*	insulin-like growth factor 2 receptor
	*INS-IGF2*	*INS*-*IGF2* read-through
	*INSR*	insulin receptor
	*RLN1*	relaxin 1
	*RLN2*	relaxin 2
	*RLN3*	relaxin 2
Kinin and tensin	*TAC1*	tachykinin precursor 1
	*TAC3*	tachykinin precursor 3
Kisspeptin	*KISS1*	KISS-1 metastasis-suppressor
	*KISS1R*	KISS-1 receptor
Motilin	*GHRL*	ghrelin and obestatin prepropeptide
	*MLN*	motilin
	*MLNR*	motilin receptor
Natriuretic factors	*NPPA*	natriuretic peptide A
	*NPPB*	natriuretic peptide B
	*NPPC*	natriuretic peptide C
Neurexophilins	*NXPH1*	neurexophilin 1
	*NXPH2*	neurexophilin 2
	*NXPH3*	neurexophilin 3
	*NXPH4*	neurexophilin 1
Neuromedins	*NMS*	neuromedin S
	*NMU*	neuromedin U
	*NMUR1*	neuromedin U receptor 1
	*NMUR2*	neuromedin U receptor 2
Neuropeptide B/W	*NPW*	neuropeptide W
	*NPBWR2*	neuropeptides Band W receptor 2
	*NPS*	neuropeptide S
	*NPSR1*	neuropeptide S receptor 1
No-family neuropeptides	*CARTPT*	CART prepropeptide
Opioid	*PDYN*	prodynorphin
	*PENK*	prokephalin
	*PNOC*	prepronociceptin
	*POMC*	propiomelanocortin
Parathyroid hormone	*PTHLH*	parathyroid hormone-like hormone
Somatostatin	*SST*	somatostatin
	*SSTR1*	Somatostatin receptor 1
	*SSTR2*	Somatostatin receptor 2
	*SSTR3*	Somatostatin receptor 3
	*SSTR4*	Somatostatin receptor 4
	*SSTR5*	Somatostatin receptor 5
Somatotrophin/prolactin	*PRL*	prolactin
	*PRLHR*	prolactin-releasing hormone receptor
	*PRLR*	prolactin receptor
Tensins and kinins	*AGT*	angiotensin
	*AGTR1*	angiotensin II receptor type 1
	*KNG1*	kininogen 1
	*NTS*	neurotensin
	*NTSR1*	neurotensin receptor 1
Thyrotropin-releasing hormone	*TRH*	thyrotropin-releasing hormone
	*TRHR*	thyrotropin-releasing hormone receptor
Vasopressin/oxytocin	*OXTR*	oxytocin receptor
	*AVP*	arginine vasopressin
	*AVPI1*	arginine vasopressin-induced 1
	*AVPR1A*	arginine vasopressin receptor 1A
	*AVPR1B*	arginine vasopressin receptor 1B

**Table 2 genes-15-00030-t002:** Demographics of the memory cohort.

Variable	Participants (n = 597) N (%)
Age group (years)	
16–25	465 (77.9)
26–35	79 (13.2)
36–45	36 (6.0)
46–55	10 (1.7)
56–65	7 (1.2)
Gender	
Male	171 (28.6)
Female	426 (71.4)
Ethnicity	
Caucasian	446 (74.7)
Other	151 (25.3)

**Table 3 genes-15-00030-t003:** Suggestive genetic associations for nine SNPs in neuropeptide-related genes with HVLT-R subcomponent performance phenotypes.

Gene	Chr	Location	SNP	Region	*β*	*t*	*p* *	Phenotype
*SCG5*	15	32965630	rs11634086	Intron	0.6918	3.47	0.0005587	Recall
*SCG5*	15	32969457	rs10519737	Intron	0.7417	3.288	0.001069	Recall
*SCG5*	15	32985954	rs1997317	Intron	0.3512	4.021	6.55 × 10^−5^	Delay
*IFG1R*	15	99449683	rs867431	Intron	−0.297	−3.438	0.0006276	Learning
*IFG1R*	15	99504437	rs2684788	3’UTR	0.0254	3.813	0.0001518	Retention
*GALR1*	18	74967894	rs9959924	Intron	0.0363	3.345	0.0008761	Retention
*OXTR*	3	8797095	rs237888	Intron	−0.347	−3.36	0.0008307	Discrimination
*CCK*	3	42299870	rs8192472	Intron	0.1716	3.312	0.0009836	Discrimination
*VIPR1*	3	42568440	rs437876	Intron	0.1755	3.332	0.0009171	Discrimination

***** Suggestive and significant *p*-value thresholds are 0.00113 and 0.0000564, respectively. *β*: beta score; *t*: T-statistic; *p*: *p*-value.

## Data Availability

All relevant data is presented within the paper and Appendix A.

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
