# Peer review of "Association Study of a Comprehensive Panel of Neuropeptide-Related Polymorphisms Suggest Potential Roles in Verbal Learning and Memory"

_genes, 2023, doi:10.3390/genes15010030_

Round 1
Reviewer 1 Report
Comments and Suggestions for Authors
In this manuscript, the authors describe a study focused on identifying potential correlations between gene variants encoding neuropeptides and mental skills in humans.
Through genetic and bioinformatic in silico research, they observed a statistical tendency associating HVLT-R test results with gene variants related to neuropeptides and neuropeptide receptors (SCG5, IGFR1, GALR1, OXTR, CCK, and VIPR1).
The authors conducted a compelling study with well-applied methods, clear objective and conclusions. The manuscript is well-written and detailed, and the experiments are qualitatively executed.
Just a few clarifications:
The authors are advised to include in the manuscript the results of statistical calculations that combine all ethnic groups into one sample. These findings would also be interesting for the readers.
Reviewer 2 Report
Comments and Suggestions for Authors
The manuscript discusses a genetic study examining the role of genetic variants in neuropeptides and neuropeptide receptors in verbal learning and memory. The research utilized the Hopkins Verbal Learning Test–Revised (HVLT-R) to assess learning and memory abilities, analyzing over 1300 SNPs present in neuropeptide and receptor genes. The mentioned neuropeptides include SCG5, IGFR1, GALR1, OXTR, CCK, and VIPR1. While the studies yielded numerous suggestive associations, they did not reach the significance threshold. The study aims to enhance understanding of the role of neuropeptides in human learning and memory, particularly in the context of genetic variations.
The topic is timely and may attract a lot of attention; however, there are numerous limitations that need to be considered before publication.
Introduction:
It would be worthwhile to expand the Introduction, providing a more detailed description of cognitive function and memory. Offer a more in-depth explanation of the brain regions and neuropeptides involved in these functions. Overall, introduce at least some of the neuropeptides examined by the authors, mention previous experiments to explore their potential roles in these functions. Provide more information on SNPs in general.
Materials and methods:
The text doesn't explicitly mention whether individuals with other neurological disorders were excluded from the study, aside from those with psychiatric disorders or head injuries. However, it's a valid point that certain neurological disorders can indeed impact cognitive and memory abilities.
The manuscript does not explicitly state whether different age groups were assessed together or separately during the evaluation of the HVLT-R test. The cohort included participants aged between 16 and 65 years. As individuals of different ages may have varying verbal learning abilities and memory capacities, considering age group differences is crucial during the assessment.
It is not entirely clear whether the genetic studies treated the data of participants from non-Caucasian ethnicities as a separate group or not.
I request the authors to provide answers to the questions/concerns I raised. It would be beneficial to supplement the manuscript with this information.
Results:
Did the authors find differences in HVLT-R, IQ, or genetic profiles among age groups, ethnic groups, or genders?
Limitation:
I did not find information on the limitations of the study that need to be taken into account when interpreting the results.
References:
In general, I recommend authors use more references to back their claims. I believe that adding more citations will help to provide better and more accurate background to this study.
Formal errors:
1. There are some spacing errors in the text.
2. For sections 2.1., 2.2., and 3.1. there is an unnecessary period in the title.
I recommend this manuscript for publication after major revision.
Comments on the Quality of English Language
There are a few areas where the language could be simplified for better readability. Moreover, there are sentences that are difficult to understand and not well-phrased, which would be worth revising.
Round 2
Reviewer 2 Report
Comments and Suggestions for Authors
Dear Authors,
I appreciate that the authors have taken my considerations into account, and all my concerns have been addressed. The quality of the manuscript has improved.